# Lateral Loaded Pile Reliability Analysis Using the Random Set Method

Marek Wyjadłowski 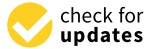

Faculty of Civil Engineering, Wrocław University of Science and Technology, 50-384 Wrocław, Poland; marek.wyjadlowski@pwr.edu.pl; Tel.: +48-713209640

**Abstract:** This study presents a procedure applied to design problems for lateral loaded piles. Calculations for a rigid concrete pile in non-cohesive soil are conducted with the aim of estimating the allowable horizontal force using the methods of Broms and Petrasovit. Random sets are applied to represent the uncertainties of soil parameters, including the internal friction angle and unit weight. Random variables are described using log-normal and beta distributions. Random set theory is utilised to represent variability in the form of probability boxes, possibility distributions, cumulative distribution functions, or intervals. Based on the assumed distributions of the subsoil, the lower and upper bounds for the precise probability of fulfilment of the limit state function of a laterally loaded pile are estimated. The reliability calculation procedure is implemented using the R package (R Studio v2024.12.1+563), and the limit forces and reliability indicators calculated using the two considered methods are compared. The presented procedure serves as an example of the use of a probabilistic approach for the assessment of the capacity of a laterally loaded pile, using a setup for the task involving set-based data and discrete probability distributions.

**Keywords:** lateral loaded pile; reliability of structures; random set method; soil uncertainties

## 1. Introduction

Pile foundations are geotechnical structures which are commonly used when a large foundation depth is necessary to transfer the effects of the superstructure to the load-bearing soil layers [1]. The piles transfer loads to these load-bearing geotechnical layers, which lie much deeper than the lowest floors or structural elements, the depth of which is determined according to the function of the building or structure.

Pile foundations are widely used for structures that are not only subject to vertical loads from the superstructure, but also to variable lateral loads (e.g., wind, waves, earthquakes) or other constant horizontal or diagonal loads (e.g., soil pressure). For example, piles serving as structural elements of sheet pile walls and bridge foundations usually operate under complex load conditions, especially longitudinal and horizontal forces, while in the case of deep excavations, horizontal forces dominate. Foundations transmitting a combination of vertical and horizontal forces to the ground are designed using raker piles. Pile technologies include the use of various construction materials, such as concrete, steel, and cemented soil. Each material is chosen for use according to its strength properties and the load that it will be subjected to. In structures subjected to impacts such as cyclic thermal loading [1], axial load and thermal movements [2], and thermal-induced flexural strain cycles leading to low-cycle fatigue [3], steel H-piles can be used effectively. This type of foundation is widespread, being particularly integral to concrete bridges. It should be

noted that, in the foundations of bridge abutments, the rotation of the pile head and the lateral movement of the pile in the direction of the backfill differ from those in the opposite direction due to the restraining effect of the backfill [4]. It is also worth mentioning the use of piles in stabilising slopes for the prevention of landslides. To reduce the impact on the environment, technologies such as deep soil mixing or fibre-reinforced concrete are used. In this type of application, the piles bear the soil pressure that may initiate a landslide, which can be considered a fixed load. Various methods for soil improvement, such as deep cement mixing (DCM) and the innovative technique called the T-shaped DCM (TDM) column, have been designed to reinforce slopes and communication embankments on soft soil, limiting their settlement [5]. A previous study [6] presented a reliability-based settlement analysis of columns using the above-mentioned technologies. Individual and system failure probabilities were analysed in terms of serviceability limit state (SLS) requirements. For the reliability analysis and to estimate the probability of failure, the Adaptive Kriging Monte Carlo Simulation was conducted.

The reliability analysis is justified, as serious failures have been reported due to lateral impacts on piles [7,8]. The prediction of the bearing capacity of piles is a complex task, as its value is a function of various factors, such as those associated with pile technology, concreting technology, the traits of concrete, soil conditions, and pile geometry. Well-grounded pile design approaches require in-depth geotechnical investigations, including planned site investigations, as well as field and laboratory testing. The results of geotechnical research influence the selection of pile technology. Due to the uncertainties involved in the design of piles, it is difficult to accurately predict the pile capacity, even when conducting sophisticated analyses.

The methods for determining the ultimate load capacity of piles can be generally categorised into several groups. In particular, these methods [9,10] include the following:

- the results of static load tests, which are consistent with other relevant methods;
- empirical or analytical calculation methods that have been validated through static load tests under comparable conditions;
- the results of dynamic load tests, whose validity has been confirmed through static load tests;
- the observed load-bearing capacity of a comparable pile foundation, provided that this approach is supported by field and soil test investigations.

The above methods can be utilised for the estimation of the ultimate load-bearing capacity, which provides a basis for calculating the characteristic and design load-bearing capacities of a pile. The assessment of the limit load capacity of piles on the basis of the test loads is essential, as it has been previously indicated [9] that this process serves as the basis for the design of piles. However, at the same time, a procedure for interpreting the results of the test loads in order to assess this value was not provided. In view of technical and time constraints, the load on the pile that causes failure during the load test is often not achieved. Therefore, the ultimate load capacity is not always determined directly from the load test, but on the basis of a selected extrapolation method. A comparison of the methods used in selected standards was presented in a previous study [11]. The assessment of load-bearing capacity based on static load tests is also subject to uncertainty resulting from the accuracy of geodetic methods [12]. The displacements of piles loaded with soil pressure working in a sheet pile wall have been examined along the entire pile using inclinometer measurements [13], which were primarily utilised to assess the serviceability limit state. Due to the above-mentioned limitations related to estimating the ultimate load capacity based on load tests, analytical methods are still important in design practice. According to ref. [14], three basic conditions—the ultimate limit state (ULS), serviceability limit state (SLS), and economic requirements limiting financial outlays—should be met. To

ensure the implementation of design assumptions and to create functional and effective pile foundations, both theory and the use of a parametric design process are necessary, which takes into account uncertainties in the parameters of the soil–structure system. The traditional deterministic approach to design uses existing experience and engineering intuition. A structure designed in accordance with standard procedures may still fail with a certain probability [15]. The soil medium has random features, and in relation to the construction material, the randomness of its parameters affects the operational phase, with environmental influences potentially degrading the strength parameters of the pile [16]. The traditional design principle (i.e., strict application of normative guidelines), as a deterministic approach in relation to geotechnical problems, does not guarantee the safety of the structure–soil system. Therefore, for complex structural systems in the ground, it is recommended to use a probabilistic approach. The development of structural design methods using reliability theory can be seen as a response to economic and efficiency requirements. The evolution of such approaches has consisted of formulating probabilistic approaches for design parameters, relying on the statistical analysis of empirical research results [17,18].

This article presents an example of the use of a probabilistic approach for the assessment of limit states in cases where the data are provided in the form of set-based information and discrete probability distributions.

## 2. Materials and Methods

### 2.1. Methods for Estimation of Lateral Pile Load Capacity

In the field of geotechnical engineering, the modelling of laterally loaded piles is one of the most complicated problems involving soil–structure interactions. Modelling the laterally loaded pile is complex due to the occurrence of many non-linearities. First, the soil stiffness is non-linear. In the case of small deformations, the soil medium exhibits greater stiffness than in the case of larger deformations. The maximum strength of the soil medium and its stiffness increase non-linearly with depth and also varies non-linearly with the pile diameter. The distribution of soil resistance values along the pile is also non-linear.

There are a large number of models available in the geotechnical engineering field that can be used to design laterally top-loaded piles [19]:

- Blum's model [20];
- Brinch Hansen model [21];
- Broms model [22];
- Characteristic Load Method (CLM) [23];
- Non-dimensional Method [24];
- p-y Curves [24];
- 3D finite element method [25].

According to refs. [14,26], the structure should meet three basic conditions: the ultimate limit state (ULS), the serviceability limit state (SLS), and economic requirements. The lateral load capacity of piles can be considered from two different aspects: the allowable and ultimate load capacities. The estimation of the allowable lateral load capacity $H_{all}$ requires displacement calculations, whereas the ultimate lateral load capacity $H_u$ is connected with the strengths of the substrate and/or pile.

The abovementioned theoretical methods were developed as a result of a theoretical analysis of the pressure imposed by the soil on rigid retaining walls. The ultimate lateral resistance per unit width of a rigid pile is greater than that of an equivalent retaining wall, due to the shear resistance on the vertical sides of the failure surfaces in the ground. If the same distribution of the ultimate soil pressure on the pile is assumed for the wall, the three-dimensional nature of the loaded pile should be taken into account by multiplying

the net soil pressure on the wall by a shape coefficient. To calculate the ultimate lateral load capacity $H_u$, it is necessary to know two basic quantities: the value of the lateral soil resistance $p_u$ for a given soil condition and the distribution of the lateral soil pressure $p_z$ with respect to the pile embedding depth. The lateral soil resistance $p_u$ reflects the unit load capacity of the circumfluent subsoil and is usually defined in the fully passive state. Meanwhile, the lateral soil pressure $p_z$ constitutes a certain ratio of $p_u$, which varies with depth.

Common methods to estimate these components, based on experimental test results, have been described in the works of Brinch Hansen [21], Broms [22], Petrasovit and Award [27], and Prasad and Chari [28]. Figure 1 depicts the assumptions of two of the abovementioned methods.

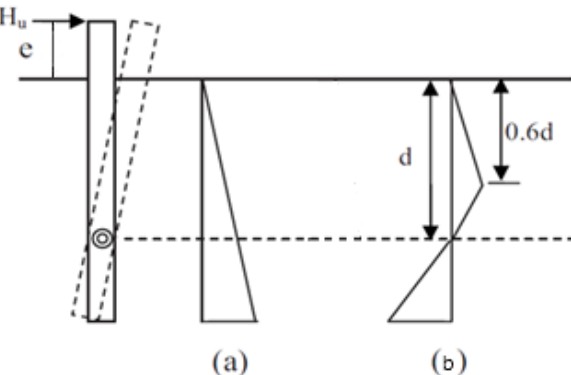

**Figure 1.** Distribution of lateral soil pressure at ultimate state: (**a**) Broms method and (**b**) Petrasovit method.

These methods assume some alignment of the lateral soil pressure distribution along the length of the pile, which is considered to be linear along the width of the pile. From the different theories presented in Figure 1, it can be generally stated that the pressure $p_u$ at any depth $z$ is given by the equation:

$$p_u = sf(\varphi)\gamma z = sf(\varphi)\sigma'_v \tag{1}$$

where

y is the density of the soil;
$\sigma'_v$ is the vertical effective stress;
$f(\varphi)$ is a function of the internal friction angle of the soil;
s is the shape factor.

The notation $p_u$ represents the peak soil passive pressure at any depth $z$ below ground level. In the Broms method for rigid piles, it is assumed that a pile rotates according to the pile base and the lateral soil resistance $p_u$ is completely mobilised throughout the entire depth of the pile (i.e., $p_z = p_u$), as shown in Figure 1. The lateral soil resistance $p_u$ in the method detailed in ref. [22] is defined as follows:

$$p_u = 3 \cdot K_p \cdot \sigma'_v \tag{2}$$

According to the earth pressure theory, Rankine's passive stress $\sigma'_p$ is equal to $K_p$ multiplied by $\sigma'_v$. For laterally loaded piles, however, the lateral soil resistance $p_u$ is greater than $\sigma'_p$ due to the three-dimensional characteristics of the lateral soil resistance [29,30]. The

ultimate lateral load capacity $H_u$ is obtained from the equilibrium condition of the resultant forces acting on the pile:

$$H_u = \frac{K_p \cdot \sigma'_{v,b} \cdot L^2 \cdot D}{2(e+L)} \tag{3}$$

The differences in assumptions about the shape coefficient value should be noted; the shape factor is represented by s, which is set to 3.0 in the Broms method and 3.7 in the Petrasovit and Award method. Detailed assumptions and derivations of the limit force values $H_u$ have been presented in ref. [27]:

$$x = \left[ -(0.567L + 2.7e) + \left( 5.307L^2 + 7.29e^2 + 10.541eL \right)^{0.5} \right] / 2.1996 \tag{4}$$

$$H_u = 0.24 \left[ 10^{1.3tan\varphi + 0.3} \right] \gamma x B [2.7x - 1.7L]. \tag{5}$$

The following notation has been adopted in the equations: x, point of rotation; e, eccentricity of loading. It should be noted that the existing methods for estimating the ultimate horizontal force are appropriate only for one layer and uniform soil conditions, resulting in the linear variability of the lateral soil resistance $p_u$ with depth, and for free-head short rigid piles in granular soils. These classical methods were chosen for further calculations due to the different locations of the pile's rotation centre assumed in these methods. In particular, in this article, the Broms method and the Petrasovit and Award method for short rigid piles are used in combination with the random set method.

*2.2. Reliability Measures*

The full range of uncertainty and its dual nature can be defined [29] in terms of the following:

-   Aleatoric Uncertainty: uncertainty which results from the fact that a parameter can behave in random ways (stochastic, objective uncertainty);
-   Epistemic Uncertainty: uncertainty which results from the lack of knowledge about a parameter (state of knowledge, subjective uncertainty or unawareness).
-   Uncertainties in geotechnical properties can be classified into three categories:
-   inherent variability;
-   in situ measurement;
-   transformation model.

These are uncertainties that exist in the process of determining the parameters of the native soil, relating to the case under consideration. For the purposes of reliability calculations, a limit state function must be defined, which takes the following general form:

$$g(x) = \begin{cases} \geq 0 \; for \; safe \; states \\ < 0 \; for \; failure \; states \end{cases}, \tag{6}$$

where $x$ is a vector of basic random variables $X = (X_1, X_2 \dots, X_n)$.

The vector coordinates are random variables representing the parameters of the structure–soil system; in the considered case, the pile–soil medium system. The basic reliability concept—the margin of safety—is given by the following expression, where $M$ denotes the safety margin:

$$M = E\{g(X)\} \tag{7}$$

where

$E\{g(X)\}$ is the expected value of the random variable $g(x)$;
$g$ is the limit state function.

By writing the limit state function in the form

$$g = R - S \tag{8}$$

based on the equations above, we obtain the following:

$$M = E\{R\} - E\{S\}. \tag{9}$$

The probability of failure, interpreted as exceeding the above condition, is as follows:

$$p_f = P\{g(X) \leq 0\} = \int\limits_{\{g(x) \leq 0\}} dF_x(x) \tag{10}$$

The simplest design principle is the strict application of normative guidelines; such a solution to the problem is called a deterministic approach. A continuation of these considerations is the design of structures that take into account probabilistic methods. In the particular context of pile design, uncertainties relate to the unknown properties of the ground, the soil–structure interactions, and pile geometry and strength parameters.

Quantifiable expressions of uncertainty (failure/non-performance probabilities or structural reliability) can be represented using four analysis methods in the related literature and design standards [31]:

- Level I: Uncertain parameters are modelled using one single nominal value (e.g., characteristic values with partial safety factor design);
- Level II: Uncertain parameters are implicitly modelled with a normal distribution according to two values (e.g., mean and variance or characteristic value). These are considered as semi-probabilistic methods;
- Level III: Uncertain quantities are modelled according to their distribution functions and correlations, leading to a calculated failure probability (e.g., full probabilistic Monte Carlo simulations);
- Risk-based methods: The consequences of failure are also considered as part of the design criteria.

An evaluation of some of these procedures was conducted in the study [19]. For non-homogeneous soils, the application of conventional methods is complex due to the non-linear variation of soil properties with depth.

### 2.3. Basic Concepts of Random Set Method

The procedure presented in this study is an example of the use of a probabilistic approach for the assessment of the laterally loaded pile capacity $H_u$, using a setup for the task that includes set-based data and discrete probability distributions. In particular, random sets are employed to describe the variability of subsoil parameters or the geometric properties of the pile.

The concept of a random set, in the form of a region that is dependent on chance, appeared in ref. [32], and an overview of the modern theory of random sets has been provided in ref. [33].

The development of the mathematical theory of random sets can be traced to the work in ref. [34]. The main new feature in modern random set theory is that random sets can have different shapes, and the progress of this concept is important for the study of random sets.

Classical random set theory describes random closed sets. The key point regarding this approach is that random points (i.e., random sets that are singletons) are closed, and so,

closed random set theory considers the original case of random points or random vectors as an exception.

Let $X$ be a non-empty set containing all possible values of a variable $x$. A random set on $X$ is written as a pair $(\Im, m)$, where $\Im = \{A_i : i = 1, \ldots, n\}$, and $m$ is a mapping $m : \Im \rightarrow [0, 1]$, to which the following conditions apply:

$$\sum_{A \in \Im} m(A) = 1$$
$$m() = 0. \tag{11}$$

$\Im$ is named the support of the random set, the sets $A_i$ are the focal elements (where $A_i \subseteq X$), and $m$ is named the basic probability assignment. Each set $A \in \Im$ includes possible values of the variable $x$, and $m(A)$ is the probability that $A$ is within the value range of $x$. In view of this imprecise definition, we are unable to estimate the accurate probability *Pro* of a generic $x \in X$ or generic subset $E \subset X$, and can only derive lower and upper bounds on these probabilities:

$$Bel(E) \leq Pro(E) \leq Pl(E),$$
$$\text{Bel}(x) \leq Pro(x) \leq Pl(x) \tag{12}$$

For a particular case, when $\Im$ includes exclusively single values (singletons), then $Bel(E) = Pro(E) = Pl(E)$ and $m$ is a probability distribution function. In the work [35], the lower bound *Bel* and the upper bound *Pl* of the probability measure were specified for each subset $E \subset X$:

$$Bel(E) = \sum_{A_i:\ A_i \subset\ E} m(A_i),$$
$$Pl(E) = \sum_{A_i:\ A_i \cap E\ \neq} m(A_i), \tag{13}$$

where the belief function *Bel* of a subset $E$ is a set-evaluated function obtained through the summation of basic probability assignments for the subsets $A_i$ included in $E$, and the plausibility function *Pl* is a set-evaluated function obtained through the summation of basic probability assignments for subsets $A_i$ that have a common area with $E$. These functions cover all reachable cumulative distribution functions that are coherent with random variables.

### 2.4. The Bounds on the Functional Response

The presented basic concept for determining the upper and lower probability bounds for generic $x \in X$ or $E \subset X$ is adopted to set the limits of the value of a function, which is an image of the arguments (also presented as random sets).

Random set theory provides a mathematical framework for linking probabilistic and set-based data, where the expansion of random sets through a functional relation is a straightforward process [36]. If the function $f$ is a mapping on $X_1 \times \ldots \times X_n$ and $x, \ldots x_n$ are variables whose values are unknown, then the deficient knowledge regarding $\mathbf{x} = (x, \ldots x_n)$ can be explained as a random relation $R$, which is a random set $(\Im, m)$ on the Cartesian product $X_1 \times \ldots \times X_n$. The new random set $(\mathfrak{N}, \rho)$, which is a projection of the function $f$, is described by the formula:

$$\mathfrak{N} = \{R_j = f(A_i),\ A_i \in \Im\} : f(A_i) = \{f(\mathbf{x}), \mathbf{x} \in\ A_i\}, \rho(R_i) = \sum_{A_i:R_j=f(A_i)} m(A_i) \tag{14}$$

If $A_1, \ldots A_n$ are sets on $X_1 \times \ldots \times X_n$, and $x, \ldots x_n$ are an independent random set, then the joint basic probability of the product is defined using the formula:

$$m(A_i \times \ldots \times A_n) = \prod_{i=1}^{n} m_i(A_i), \; A_i \times \ldots A_n \in R. \tag{15}$$

The result of the function is allocated a probability, which is the product of the probabilities that have been allocated to the sets that are the arguments of the function. If the focal set $A_i$ is a fixed interval of real numbers—that is, $A_i = \{x : x \in [l_i, u_i]\}$—then the lower and upper cumulative distribution functions at point $x$ are accordingly given by the following:

$$F_*(x) = \sum_{i: x \leq u_i} m(A_i) F^*(x) = \sum_{i: x \geq l_i} m(A_i). \tag{16}$$

Figure 2 presents the structure of a random set originating from various data given as intervals (i.e., focal elements $A_1, \ldots A_n$ and their basic probability assignments $m_1, \ldots m_n$). To draw the appropriate right border, the probability mass for each subset is assumed to be focused at the upper bound of the subset [37].

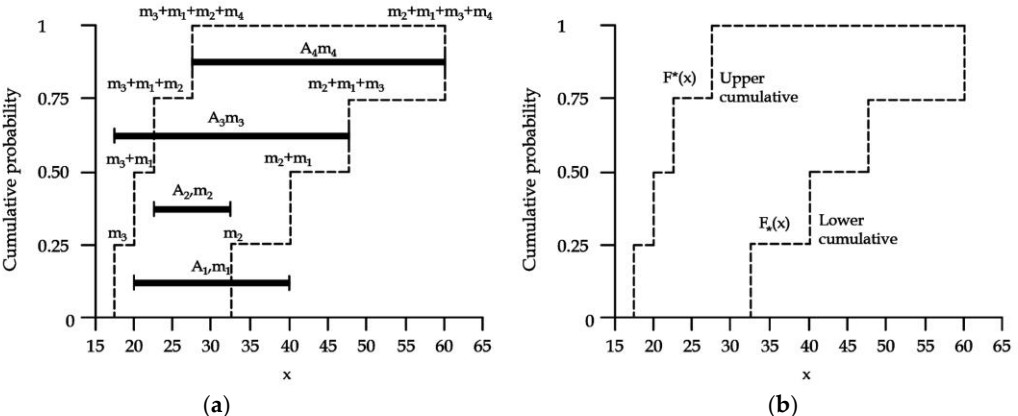

**Figure 2.** Graphical representation of the random set method: (**a**) construction of the random set and (**b**) upper and lower discrete cumulative distribution functions.

To obtain the left border, the distribution of the probability mass of each subset in the calculation matrix is assumed to be focused at the lower bound of the subset.

*2.5. Reliability Problem Using Random Set Theory*

The procedure for determining the lower and upper cumulative distribution functions of the functional response is of general use and can be used for probabilistic tasks when the functional response of the limit state function is sought. The use of random set theory involves the calculation of the bounds on $p_f = P\{g(X) \leq 0\}$, subject to the permitted values contained in the random subsets.

Let $F \subseteq X$ be a set of failed states (i.e., $g(X) \leq 0$) of the limit state function, described in terms of the functional response. The plausibility $Pl(F)$ and belief $Bel\,(F)$ functions are given as follows:

$$Bel(F) = \sum_{A_i: \, A_i \subset F} m(A_i),$$
$$Pl(F) = \sum_{A_i: \, A_i \cap F \neq} m(A_i). \tag{17}$$

Then, the upper and lower bounds for the probability of failure are given by the following:

$$Bel(F) \leq p_f \leq Pl(F). \tag{18}$$

The application of the proposed method to solve practical boundary value problems is presented on the basis of the horizontally loaded pile task in the following section.

## 3. Application and Results

This section is divided into four main parts, as follows:

1.  Description of the pile computational model;
2.  Considerations regarding the results of deterministic classical calculation methods;
3.  Summary of results using the probability distribution functions calculated according to random set theory;
4.  Presentation of results of reliability calculations.

### 3.1. Selecting a Pile Model for Calculations

The soil–structure interaction task was taken into consideration, especially for concrete piles with a diameter of D = 0.36 m in non-cohesive soils, which, for example, corresponds to bored piles made using FDP technology [38]. Advantages of FDP technology include increased shaft friction and bottom resistance due to the displacement of the soil surrounding the pile [39]. Before starting the calculations, it should be confirmed that the assumed piles can be considered rigid, as required for the Broms method [22]. Laterally loaded piles are generally considered rigid, based on the value of $K_{rs}$:

$$K_{rs} = E_p I_p / E_h L^4 \qquad (19)$$

where:

-   $E_p$ is the elastic modulus of the pile material (for C25/30 class concrete, $E_p$ = 31.47 GPa);
-   $I_p$ is the moment of inertia of the pile (at a diameter D = 0.38 m, $I_p = 1.04^{-3}$ m$^4$);
-   $E_h$ is the horizontal soil modulus at the pile's base (9.90 MPa, 16.10 MPa, and 23.90 MPa for density index $I_D$ values of 0.25, 0.50, and 0.70, respectively; this covers the entire considered range of friction angle variability);
-   $L$ is the embedded length of the pile (assumed to be $L$ = 6.0 m in this study).

For a rigid pile, the value of the relative stiffness factor $K_{rs}$ should meet the condition $K_{rs} > 10^{-1}$ to $10^{-2}$, depending on the degree of fixity of the pile head [40].

For the assumed pile and substrate parameters, the relative stiffness factor took values of $K_{rs}$ = 0.025, 0.015, and 0.011 at density index values $I_D$ of 0.25, 0.50, and 0.70, respectively, indicating that all concrete piles can be considered rigid.

### 3.2. Generation of Data Sets

Calculations of the limit horizontal force were carried out for the two previously discussed models and compatible random sets. The soil was considered a homogenous isotropic layer. The basic variables for the random set model were the internal friction angle $\varphi'$ and unit weight $\gamma$. The subsoil parameters which were applied as basic variables are summarised in Table 1.

**Table 1.** Basic variables for material parameters (input values).

| Soil, Information Source | $\varphi'$ [°] | $\gamma$ [kN/m$^3$] |
|---|---|---|
| Geotechnical report | 29.0–36.0 | 17.0–19.0 |

The characteristics of random variables for the soil layer, which were treated as basic variables, are summarised in Table 2.

**Table 2.** Characteristics of random variables.

| Medium Sand | $\varphi'$ [°] | $\gamma$ [kN/m$^3$] |
|---|---|---|
| Mean value | 33.20 | 18.00 |
| Standard deviation | 1.53 | 1.80 |

Table 1 highlights the wide range of parameters considered, which reflects the technical judgement of those involved and the geotechnical variability. After taking the average input parameters, the next step is to define the input random sets. The focal elements were directly determined from the intervals defined by the values computed from the probability distributions. With the values calculated from the probability distribution functions for each stochastic input parameter (included in Table 1) and their corresponding probabilities, the random set was completely defined. In Table 3, the four basic probability assignments $m$ correspond to the probabilities associated with every focal element that belongs to the random set on $X$. Here, these random sets are defined based on the mean parameters calculated above and the two different probability distributions. The random set which represents the internal friction angle is defined by a pair $(\Im^\varphi, m^\varphi)$, where $\Im^\varphi$ is the support of the random set and $m^\varphi$ is the basic probability assignment. The support $\Im^\varphi$ consists of the following focal elements:

$$\Im^\varphi = \left\{ A_i^\varphi : i = 1, \ldots, 4 \right\}. \tag{20}$$

**Table 3.** Details of random set computed according to log-normal distribution.

| Random Variable | Probability of a Value Occurring | Focal Elements $A_i$ | | Basic Probability Assignment $m_i$ |
|---|---|---|---|---|
| | | Lower Bound | Upper Bound | |
| | $1.0 \times 10^{-5}$ | 27.358 | 30.654 | 0.05 |
| | 0.05 | 30.654 | 33.220 | 0.45 |
| Friction angle $\varphi'$ | 0.50 | 33.220 | 36.675 | 0.45 |
| | 0.95 | 36.675 | 38.469 | 0.05 |
| | 0.99 | 38.469 | | |

The focal elements were directly determined from the intervals defined by the values computed according to the probability distributions, using the R package.

The probability distributions for the considered variables are presented in Figures 3 and 4.

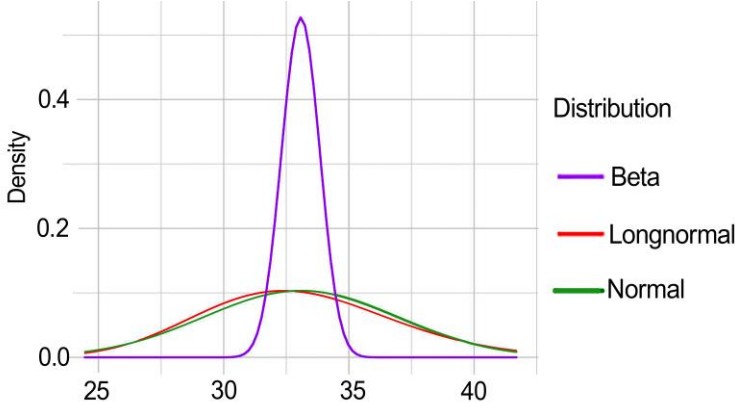

**Figure 3.** Probability distributions for the internal friction angle variable.

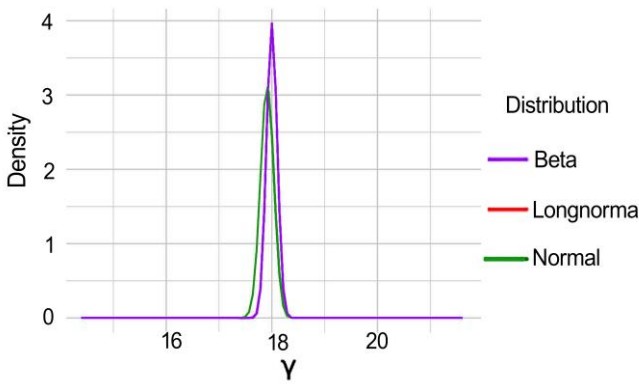

**Figure 4.** Probability distributions for the soil unit weight variable.

The values selected according to these probabilities for the friction angle parameter are summarised in Table 3 (for log-normal distribution) and Table 4 (for beta distribution).

**Table 4.** Details of random set computed according to beta distribution.

| Random Variable | Probability of a Value Occurring | Focal Elements $A_i$ | | Basic Probability Assignment $m_i$ |
|---|---|---|---|---|
| | | Lower Bound | Upper Bound | |
| | $1.0 \times 10^{-5}$ | 17.40 | 17.80 | 0.05 |
| | 0.05 | 17.80 | 18.00 | 0.45 |
| Unit weight $\gamma$ | 0.50 | 18.00 | 18.20 | 0.45 |
| | 0.95 | 18.20 | 18.40 | 0.05 |
| | 0.99 | 18.40 | | |

*3.3. Results of the Deterministic Calculation*

The basis for the implementation was two classical pile analysis methods presented in the previous section, and calculations were conducted for various possible combinations of input parameters. First, the results regarding the limit values of horizontal force under the selected geotechnical parameters are presented in Table 5, with the assumption of a deterministic nature of the substrate. The presented values provide insight into the possible range of results according to the assumed random variables and calculation method.

**Table 5.** Ultimate limit force for selected deterministic values of geotechnical parameters.

| Calculation Method | $\varphi'$ [°] | $\gamma$ [kN/m³] | $H_u$ [kN] | Remark |
|---|---|---|---|---|
| | 30.0 | 17.0 | 349 | min $H_u$ |
| | 33.5 | 18.0 | 423 | |
| Broms | 35.0 | 18.0 | 453 | mean $H_u$ |
| | 36.5 | 18.0 | 487 | |
| | 40.0 | 19.0 | 596 | max $H_u$ |
| | 30.0 | 17.0 | 214 | min $H_u$ |
| | 33.5 | 18.0 | 288 | |
| Petrasovit | 35.0 | 18.0 | 326 | mean $H_u$ |
| | 36.5 | 18.0 | 373 | |
| | 40.0 | 19.0 | 522 | max $H_u$ |

The table presents the minimum, average, and maximum values of the limit force that can be obtained within the range of the assumed parameters.

### 3.4. Calculation of Probability Mass Functions Based on Random Set Theory

In order to generate additional data for the reliability analysis, a series of combinations of random sets were generated according to both log-normal and beta distributions; these combinations are detailed in Table 6. Regarding the probability mass functions obtained from these combinations of random sets, it is worth mentioning the following:

- the same combinations of random sets were used for the two calculation methods;
- the influence of one parameter and two geotechnical parameters was considered in combinations of random sets.

**Table 6.** Combination of random sets, calculated according to different probability distributions.

| | Parameter | | Model | Combination Data |
| | Friction Angle $\varphi'$ | Unit Weight $\gamma$ | | |
|---|---|---|---|---|
| Distribution | Log-normal | - | Broms | 1B |
| | Log-normal | beta | | 2B |
| | Log-normal | - | Petrasovit | 1P |
| | Log-normal | beta | | 2P |

The probability mass function derived from each parameter combination is shown in Figure 3. From these functions, the data for the reliability assessment were extracted. The considered random sets and the probability assignments for the input parameters are detailed in Tables 4 and 5. The selection of these random sets was carried out in order to assess the impacts of extreme values on both the safe and extreme sides of the calculations, as well as to give more weight to intervals close to the mean of the distribution, given that these are the most probable parameters. Therefore, the first and fourth focal elements were assigned a probability of 0.05, while the second and third were assigned a probability of 0.45 (see Table 4).

The number of calculations $n_c$ required to find the bounds on the system response is presented in the following Equation (21):

$$n_c = 2^N \prod_{i=1}^{N} n_i, \tag{21}$$

where

N is the number of basic variables;
n is the number of information sources available for each observation.

The number of calculations can be reduced if the function is exactly monotonic, which is a valid assumption for all analyses discussed. In this case, the vertices on which the lower and upper bounds lie in the random set can be identified simply by considering the direction of growth of ultimate lateral load $H_u(A_i)$, which can be determined through a sensitivity analysis. Thus, $H_u(A_i)$ only needs to be calculated twice for each focal element $A_i$.

Figure 5 shows probability distribution calculated according to log-normal distribution of internal friction.

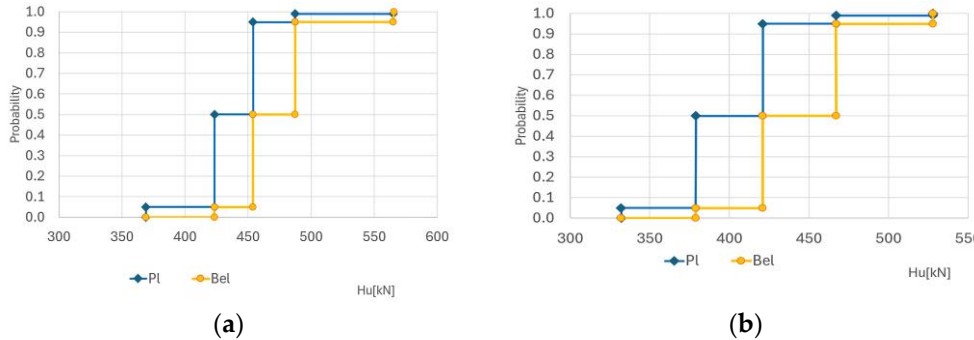

**Figure 5.** Probability distribution function envelope for ultimate horizontal load. Combination data from: (**a**) Broms method (1B) and (**b**) Petrasovit method (1P).

### 3.5. Focal Elements Calculated According to Both Log-Normal and Beta Distributions

The basic framework is the simplest case, where one random variable with log-normal distribution was analysed. Thus, the function $H_u(A_i)$—which represents a numerical model in this framework—must be evaluated $2^N$ times for each focal element $A_i$. Regarding the number of calculations, there is one random variable and four focal elements that consist of an upper and lower bound. Thus, the model was calculated a total of 16 times: 8 times for the upper bound and 8 times for the lower bound of the probability distribution function. The probability distributions for the considered methods are presented in Figure 6.

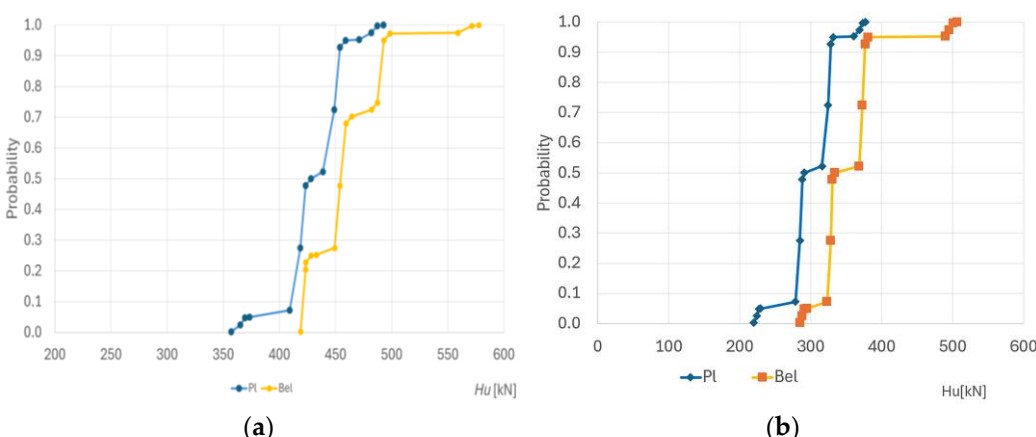

**Figure 6.** Probability distribution function envelope for ultimate horizontal load. Random set calculated according to log-normal and beta distributions. Combination data from: (**a**) Broms method (2B) and (**b**) Petrasovit method (2P).

### 3.6. Presentation of Results of Reliability Calculations

The reliability evaluation was carried out based on the limit state function g(x), as previously defined in Equation (6). The limit state function for a horizontally loaded pile can be defined in terms of the ultimate limit load of the pile as follows:

$$g(\boldsymbol{x}) = H_u^* - H_u\left(X_\varphi, X_\gamma\right), \tag{22}$$

where

$H_u^*$ is a measurement of the allowable load at the head of the pile, selected arbitrarily;
$H_u\left(X_\varphi, X_\gamma\right)$ is a measurement of the ultimate horizontally load (which is a function of the random variables internal friction angle and unit weight, respectively).

The reliability is then the probability $p = P\{g(X) > 0\}$ and, therefore, is the complement of the probability of failure, as given by the equation:

$$p = 1 - p_f. \tag{23}$$

Utilising random set theory, the reliability problem is reduced to an evaluation of the bounds on $p_f = \mathrm{P}(\mathrm{g} \leq 0)$ subject to the available knowledge restricting the values of the input parameters. If the set of failed states is labelled $F \subset X$ (as g is a function of X), the upper and lower bounds on the probability are the plausibility and belief functions, respectively:

$$Bel(F) \leq p_f \leq Pl(F). \tag{24}$$

Based on the determined load capacity envelopes, the reliability assessment and the envelopes of reliability indices were estimated, see Figure 7.

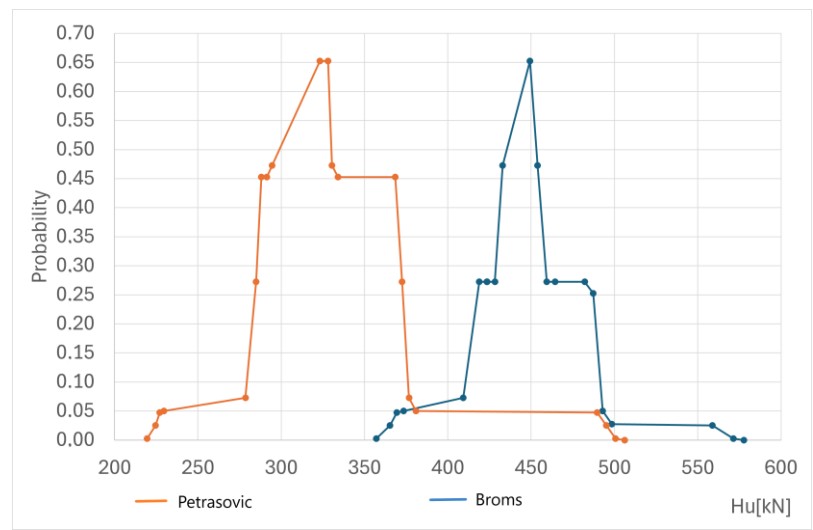

**Figure 7.** Estimated interval of failure for calculation methods: Broms method (2B) and Petrasovit method (2P).

The reliability indices for the two pile calculation methods are presented in Figure 8.

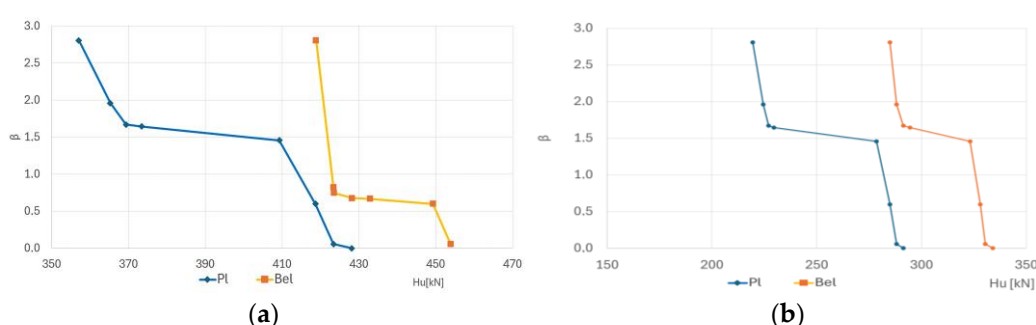

**Figure 8.** Reliability indices: (**a**) Broms method (2B) and (**b**) Petrasovit method (2P).

Numerical reliability values are often introduced on the basis of the reliability index $\beta$, defined by the following:

$$\beta = -\phi^{-1}\left(p_f\right), \tag{25}$$

where $\phi^{-1}$ is the inverse of the standard normal cumulative distribution.

The abovementioned function returns the inverse of the cumulative standard normal distribution, where this distribution has a mean of zero and a standard deviation of one.

### 3.7. Calculation of Probability Mass Functions with Scouring Effect

Scour denotes a phenomenon in which soils around foundations are removed by currents and waves. The scouring effect is acknowledged as a primary contributor to the weakening of the bearing performance of bridge foundation piles, and asymmetric scour patterns around the pile often occur [41,42]. This phenomenon is gaining importance in research focused on offshore monopiles [43] and landslide [44] scenarios. The ultimate load-bearing capacity of piles is influenced by the geotechnical parameters of the soil medium, as considered in the previous examples, and by the length of the pile. The presented calculations assume a constant scour depth, and the lower part of the upper scour hole has a significant influence on the ultimate load-bearing capacity of a single pile. A consequence for the calculations is as follows: the horizontal action of the pile head acts according to the eccentricity e, thus reducing the immersion into the substrate L. However, the pile can still be considered rigid. The two considered calculation methods take into account such changes in the task parameters.

The length of the pile is described by a two-point distribution. For the pile length parameter, two focal elements with a probability of occurrence of 50% were assumed: the embedded pile length $L = 5.50$ m under the scouring effect and the pile length without the scouring effect. The model contains four focal elements for the internal friction angle, four elements for the unit weight of the soil, and two focal elements for the pile length (i.e., the design length and the shortened length). The number of calculations necessary to build the envelope is $4 \times 4 \times 2 = 32$ calculations for each envelope. The calculation results for the two task models used are presented in Table 7 and Figure 9. This probability calculation is an extension of the previous example, including an altered pile length and eccentricity load effect.

**Table 7.** Combinations of random sets for the analysis of the scour effect.

| | Parameter | | Model | Combination Data |
|---|---|---|---|---|
| | Friction Angle $\varphi'$ | Unit Weight $\gamma$ | | |
| Distribution | Log-normal | beta | Broms | 3B |
| | Log-normal | beta | Petrasovit | 3P |

**Figure 9.** Probability distribution function envelope for the ultimate horizontal load. The random set was calculated according to log-normal and beta distributions with the scour effect. Combination data from: (**a**) Broms method (3B) and (**b**) Petrasovit method (3P).

Based on the determined load capacity envelopes, the reliability assessment and the envelopes of reliability indices were estimates, see Figures 10 and 11.

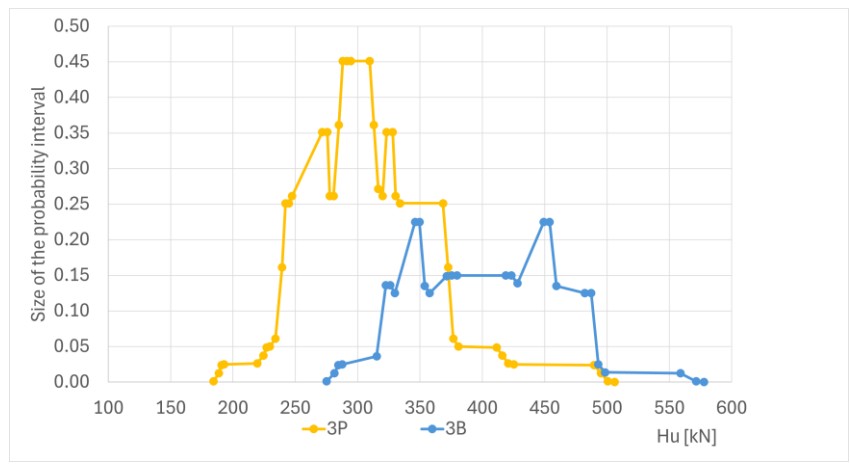

**Figure 10.** Size of the interval of failure for the Broms (3B) and Petrasovit (3P) calculation methods.

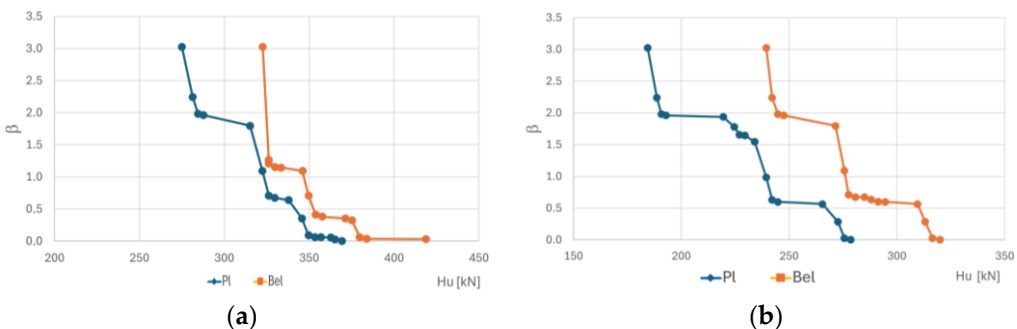

| (**a**) | (**b**) |
|---------|---------|

**Figure 11.** Reliability indices: (**a**) Broms method (3B) and (**b**) Petrasovit method (3P).

The analysis of the pile with the scour effect essentially includes the focal elements for the pile fully embedded in the subsoil, where the maximum ultimate force remains unchanged, but the minimum ultimate force obtained under this assumption is smaller.

Adding a third variable with a discrete two-point distribution requires an increase in the scope of calculations, in order to obtain bounds on the system response. The newly created calculation points are concentrated in three zones—both end zones and the central part—for which cumulative probability distribution functions are used. The shape of the obtained cumulative distribution functions is not as steep as in previous cases, appearing smoother.

The maximum values for the intervals became smaller, and there were three local maxima over the interval, instead of one (as in the previous model).

## 4. Discussion

In this work, a reliability assessment of horizontally loaded piles was carried out based on random set theory, which was performed to generate the cumulative probability mass distribution function of the ultimate horizontal load and reliability indices for a pile embedded in non-cohesive soil.

The following methods were used to describe the soil parameters:

- one random variable: the internal friction angle described by the log-normal distribution;
- two random variables: the internal friction angle described by the log-normal distribution and soil unit weight described by the beta distribution;

- three random variables: the internal friction angle described by the log-normal distribution, soil unit weight described by the beta distribution, and scouring effect described by a two-point distribution.

Estimation of the horizontal limit forces in the case of deterministic geotechnical parameters was performed using two classical models—namely, those of Broms and Petrasovit—which are limited to rigid piles in cohesionless soils. The input soil parameters were adopted based on geotechnical documentation. Random sets of geotechnical parameters were calculated based on log-normal and beta distributions with selected probability assignments, taking into account both extreme and expected input parameters along with their possible combinations.

The wide range of input parameters allowed for the calculation of a cumulative distribution function under each of the assumptions, which is extremely useful for making design decisions under both extreme and typical conditions. Regarding the calculation models developed to simulate the behaviours of laterally loaded piles, it should be highlighted that:

- the models generated results over a large range, including under extreme conditions corresponding to unfavourable combinations of soil parameters;
- the calculated values of the limit load reflect the differences in the results obtained by the models for the deterministic case.

Random set theory has been proven to be a good alternative for information intervals where there is no information related to the probability distribution between the interval boundaries, which is usually the case when geotechnical design is performed.

It should be emphasised that the forces transferred to the pile and the values considered excessive are assumed based on the information provided by the design engineers. Design to achieve the expected safety factor involves determining the allowable load that ensures that the failure probability condition is met. It is worth considering the case of a temporary structure, where the scouring effect can be neglected, and a permanent structure with the occurrence of this phenomenon. The reliability index $\beta$-values are suggested in the ISO standard [45] for the following cases:

- for fatigue limit states, use $\beta = 2.3$ to $\beta = 3.1$ depending on possibility of inspection;
- for ultimate limit states design, use the safety classes $\beta = 3.1$; 3.8; 4.3.

The allowable calculated loads $H_u$ to achieve the assumed reliability factor for temporary structures and for permanent structures are given in Table 8. Recommended high-reliability index values $\beta = 3.8$; 4.3 are not achieved for a wide range of soil parameters.

**Table 8.** Allowable $H_u$ forces for expected $\beta$-values.

| | Ultimate Load $H_u$ [kN] | | | |
| --- | --- | --- | --- | --- |
| | **Temporary Structures** | | **Permanent Structures** | |
| Broms Method | $\beta = 3.1$ | $\beta = 2.3$ | $B = 3.1$ | $\beta = 2.3$ |
| Plausibility *Pl(F)* | 350 | 360 | 270 | 280 |
| Belief *Bel(F)* | 420 | 425 | 320 | 325 |
| Petrasovit Method | $\beta = 3.1$ | $\beta = 2.3$ | $B = 3.1$ | $\beta = 2.3$ |
| Plausibility *Pl(F)* | 220 | 225 | 185 | 190 |
| Belief *Bel(F)* | 285 | 290 | 240 | 245 |

The allowable $H_u$ forces determined by two adopted methods take on different values, as is the case with deterministic calculations. These results are significantly lower than the

pile bearing capacity calculated for the average soil parameters within the range shown in Table 6. The above considerations concern the load-bearing capacity of the pile resulting from the soil parameters, but do not take into account the load-bearing capacity and reliability of the pile [46] as a structural element in the pile-soil system.

The generalised case of force acting on eccentricity was shown using the scour phenomenon as an example. The scouring effect has a more important influence on the calculated ultimate limit forces than the randomness of the unit weight. The size of the probability interval for the scour event was observed to be reduced, especially when using the Petrasovit method. Considering that the scour phenomenon occurs, the values of the limiting forces ensuring a β-index of 3 were reduced by approximately one-third with the two considered calculation methods.

The final result of the reliability analysis using the random set method does not provide an exact value of the failure probability but, instead, a reliability range between extreme cases. This is due to the fact that the method does not consider an observation itself but, instead, subsets (focal elements) in which the observation lies somewhere within the interval defined by the subset. This provides a good means of approximation for the assessment of geotechnical parameters, due to the inherent and epistemic uncertainty, making it impossible to establish exact values for the parameters. The assignment of probabilities to selected focal elements had a significant impact on the obtained forms of the cumulative distribution functions, with two opposite cases identified in this study, as explained below:

-       When focal elements consist of extreme parameters with low probability assignments and average parameters with high values of probability assignment (calculated based on log-normal distribution), discontinuous and steep probability mass functions were obtained;

-       When all the focal elements had the same probability assignment, a more continuous and shallower probability distribution was obtained.

According to the results of this research, it is more convenient to associate the same or similar basic probability assignment to the input focal elements, in order to avoid a highly discontinuous and steep cumulative probability mass function.

However, the assigned probabilities depend on the problem studied, critically relying on engineering judgement and experience regarding the most probable and extreme scenarios for each problem. In other words, the random input parameters must be bounded depending on each particular case, making appropriate decisions regarding the most probable and extreme conditions.

## 5. Conclusions

In geotechnical projects, parameters obtained from different types of tests and soil investigations are commonly used. Regarding the parameters of a pile embedded into the ground, the load eccentricity may change in different stages of the project or during its service life; as such, the probability distributions of model parameters may change. Hence, random set theory can be considered a systematic technique for dealing with this challenge, providing the engineer or expert with very useful guidance for decision-making. However, it should be remembered that the estimated probability of failure is highly dependent on the random sets and the number of sources. Therefore, the evaluation of the incoming data is very important when defining these input sets. For reliability calculations, a careful choice of the number of random variables and information sources should be made, considering that the selection of many variables increases the computational burden. The presented technique is particularly recommended for use in typical engineering applications, where little information is usually available and the probability distribution functions of the input

parameters are almost never known. Further designs and calculations can be performed for different probability density functions of the input parameters. In addition, other calculation methods for other geotechnical problems can be evaluated in order to promote the use of random sets as an efficient probabilistic technique.

**Funding:** The author would like to express their gratitude to Geohabitat P'ship for technical support of this study and donations in kind.

**Data Availability Statement:** The original contributions presented in this study are included in the article. Further inquiries can be directed to the corresponding author.

**Conflicts of Interest:** The author declares no conflicts of interest.

## Nomenclature

| | |
|---|---|
| $\gamma$ | unit weight of the soil |
| $\sigma'_v$ | vertical effective stress |
| $f(\varphi')$ | function of internal friction angle of soil |
| s | shape factor |
| $K_p$ | Rankine's passive pressure coefficient; |
| $\varphi'$ | internal friction angle of soil |
| L | embedded length of the pile |
| D | pile diameter |
| e | vertical eccentricity of lateral load from the soil surface |
| $\sigma'_{v,b}$ | vertical effective stress at pile base |
| $E\{g(X)\}$ | expected value of random variable $g(x)$ |
| g | limit state function |
| R | structural resistance |
| S | action effect |

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
