# Peer review of "Lateral Loaded Pile Reliability Analysis Using the Random Set Method"

_buildings, doi:10.3390/buildings15060882_

Round 1
Reviewer 1 Report
Comments and Suggestions for Authors
This work develops a lateral loaded pile reliability analysis with the Random Set Method. The manuscript is well structured and presents original and interesting results regarding the design problems for the lateral loaded pile. However, there are some implications that authors must address, which are presented below:
1. The size of some words on lines 287 and 288 are different from the rest of the manuscript. This also occurs when comparing eqs. 17 and 18. In this sense, the author must homogenize the text size throughout the manuscript.
2. In Table 8, the word “distribution” should be horizontal.
3. The font in the header of table 6 is different from the font used in the other tables.
4. For a better appreciation of the figures, it is important that the author consider improving the quality of all figures. The text and numbers are very small and appear blurry compared to the text in the manuscript.
5. What are the implications that cause a difference between the Broms’ method and the Petrasovit’s method? Also, under what circumstances might the results of these methods coincide?
6. To ensure the proper functioning of the methodology used in this work, the author must compare his results with any study reported in the literature.
Reviewer 2 Report
Comments and Suggestions for Authors
This study presents a procedure applied to the design problems for the lateral loaded pile. Calculations on a rigid concrete pile in noncohesive soil were aimed at estimation the allowable horizontal force using Brom’s and Petrasovit’s methods. Random sets are applied to represent the uncertainties of soil parameters: internal friction angle and unit weight. Random variables were described by a lognormal and beta distribution. Random set theory includes variability in the form of probability boxes, possibility distributions, cumulative distribution functions or intervals. Based on the assumed distributions of the subsoil the lower and upper bounds on precise probability of fulfilment of the limit state function laterally loaded pile was estimated. The reliability calculation procedure was implemented in the R package. The calculated limit forces and reliability indicators for the two methods considered were compared. In this regard, this study will be beneficial to the literature. It can be accepted after minor changes.
1. A more comprehensive introduction is necessary. Providing more references is extremely important. Especially studies in recent years should be given.
2. In what ways does this study differ from prior research conducted? This is not quite evident. Upon completion of the first section, the book must be delivered to the reader.
3.Figure 2-3-4 should be given clear.
4. According to what was said in section 3.1, it was written that L- embedded length of pile, here assumed as L = 6,0 m. why? It should be explained.
4. in section 2.3, the picture of the test set up should be added.
5. In section 2.4, it was said that “his is due to the method does not consider an observation itself but subsets (focal elements), in which the observation is somewhere within the interval defined by the subset.” Why? what was the main idea behind thid? It should be given more information.
6. It is recommended that further information, such as boundary conditions, be included in manuscript.
8. A comprehensive discussion of the findings is required. The format of a technical report is not appropriate for how presentations should be delivered.
9. More precise findings and debates need to be included in the conclusion instead. It is rather lengthy.
10. References should be updated such as “experimental and numerical studies on the proposed stiffener detailing of integral bridge steel h piles to alleviate the combined adverse effects of axial load and cyclic thermal displacements, Effect of pile orientation on the fatigue performance of jointless bridge H-piles subjected to cyclic flexural strains, Effect of thermal induced flexural strain cycles on the low cycle fatigue performance of integral bridge steel H-piles
Comments on the Quality of English LanguageThe English could be improved to more clearly express the research.
Reviewer 3 Report
Comments and Suggestions for Authors
This study presents a procedure applied to the design problems for the lateral loaded pile. 7 Calculations on a rigid concrete pile in noncohesive soil were aimed at estimation the allowable 8 horizontal force using Brom’s and Petrasovit’s methods. Random sets are applied to represent the 9 uncertainties of soil parameters: internal friction angle and unit weight. The paper is interesting, but it required revision.
1. The number of keywords is too few. I suggest authors to add at least two more.
2. Reliability-based analyses are extensively used for piles which shoul be appreciated:
l Phutthananon, C.; Jongpradist, P.; Dias, D.; Guo, X.; Jamsawang, P.; Baroth, J. Reliability-based settlement analysis of embankments over soft soils reinforced with T-shaped deep cement mixing piles. Frontiers of Structural and Civil Engineering 2022, 16(5), 638-656.
3. Problems for the lateral loaded pile are also extensively studied.
4. More attractive examples and figures are required. These are essential.
5. Please improve the English. Needs a full correction before resubmitting.
6. What is the best way to compare two Random set methods? Please discuss in the revised paper.
7. References citation should use the bracket [ ] instead of ( ).
8. Lines 44-49, 96-103, 172-175, 177-179, 209-217, 499-505, 517-521, should be written as a paragraph of description instead of a list.
9. Notations in the following equations, Lines 135-138, 145-147, 154-157, 190-191, 194-195, 309-314, 418-420, 439-440, should either give a list put in the very beginning of the article or give description after every equation. It is not proper to give a list of notations after every equation.
Comments on the Quality of English LanguageThe English could be improved to more clearly express the research.
Round 2
Reviewer 1 Report
Comments and Suggestions for Authors
The author have appropriately addressed the comments and I consider that the paper "Lateral loaded pile reliability analysis with the Random Set Method" deserves to be published in the Journal "Buildings".
Reviewer 2 Report
Comments and Suggestions for Authors
all requirements were done. it can be accepted.